



# Rethinking the correction for absorbing aerosols in the satellite-based surface UV products.

Antti Arola[1], William Wandji Nyamsi[1,2], Antti Lipponen[1], Stelios Kazadzis[3], Nickolay A. Krotkov[4], and Johanna Tamminen[5]

[1]Finnish Meteorological Institute, Kuopio, Finland.
[2]Department of Physics, Faculty of Sciences, University of Yaoundé 1, P.O. Box 812 Yaoundé, Cameroon.
[3]Physikalisch-Meteorologisches Observatorium Davos, World Radiation Center (PMOD-WRC), Davos, Switzerland.
[4]NASA Goddard Space Flight Center, Greenbelt, MD, USA.
[5]Finnish Meteorological Institute, Helsinki, Finland.

**Correspondence:** Antti Arola (antti.arola@fmi.fi)

**Abstract.**

Satellite estimates of surface UV irradiance have been available since 1978 from TOMS UV spectrometer and continued with significantly improved ground resolution using Ozone Monitoring Instrument (OMI 2004-current) and Sentinel 5 Precursor (S5P 2017-current). The surface UV retrieval algorithm remains essentially the same: it first estimates the clear-sky UV irradiance based on measured ozone and then accounts for the attenuation by clouds and aerosols applying two consecutive correction factors. When estimating the total aerosol effect in surface UV irradiance, there are two major classes of aerosols to be considered: 1) aerosols that only scatter UV radiation and 2) aerosols that both scatter and absorb UV radiation. The former effect is implicitly included in the measured effective Lambertian Equivalent scene reflectivity (LER), so the scattering aerosol influence is estimated through cloud correction factor. Aerosols that absorb UV radiation attenuate the surface UV radiation more strongly than non-absorbing aerosols of the same extinction optical depth (AOD). Moreover, since these aerosols also attenuate the outgoing satellite-measured radiance, the cloud correction factor that treats these aerosols as purely scattering underestimates their AOD causing underestimation of LER and overestimation of surface UV irradiance. Therefore, for correction of aerosol absorption additional information is needed, such as the UV absorbing Aerosol Index (UVAI) or a model-based monthly climatology of aerosol absorption optical depth (AAOD). A correction for absorbing aerosols was proposed almost a decade ago and later implemented in the operational OMI and TROPOMI UV algorithms. In this study, however, we show that there is still room for an improvement to better account for the solar zenith angle dependence and non-linearity in the absorbing aerosol attenuation and as a result we propose an improved correction scheme. There are two main differences between the new proposed correction and the one that is currently operational in OMI and TROPOMI UV-algorithms. First, the currently operational correction for absorbing aerosols is a function of AAOD only, while the new correction takes additionally the solar zenith angle dependence into account. Second, the 2nd order polynomial of the new correction takes better into account the non-linearity in the correction as a function of AAOD, if compared to the currently operational one, and thus better describes the effect by absorbing aerosols over larger range of AAOD. To illustrate the potential impact of the new correction in the global UV estimates, we applied the current and new proposed correction for global fields of AAOD from the aerosol clima-



tology currently used in OMI UV algorithm, showing a typical differences of $\pm 5\%$. This new correction is easy to implement

operationally using information of solar zenith angle and existing AAOD climatology.

## 1   Introduction

Exposure to UV radiation has both beneficial and harmful effects for humans, animals and plant life (Juzeniene et al. , 2011). Human overexposure to UV has a number of negative implications, such as the acute erythema, the risk of skin cancer with accumulated UV dose and a number of eye diseases (snow blindness, cataract). On the other hand, UVB solar radiation is

linked with Vitamin D synthesis (Webb et al. , 2011). Low levels of Vitamin D are associated with a number of medical problems (Lucas et al. , 2015). Most recently also the UV radiation and risk of COVID-19 have been connected (Jain et al. , 2020; Herman et al. , 2020).

Solar UV radiation reaching the Earth's surface can be measured from the surface or can be estimated globally using satellite measurements combined with radiative transfer modeling. There are ground-based UV instruments in many places, many of

them were installed in the 90's when the ozone depletion problem became a serious environmental issue. Nevertheless, the coverage of these sites has remained notably sparse and the number of active sites has even been decreasing recently, when the issue of ozone depletion has been considered being under control. Therefore, accurate satellite estimates of surface UV are of great importance for health and biological impact studies, and the continuous global monitoring of surface UV levels is essential in the future. According to Bais et al. (2019) higher values of UV are expected by the end of 21st century relative to

the present decade for latitudes around tropics and lower values for the rest of the world.

Satellite-based UV offer the only means to obtain global coverage of surface UV estimates. Surface UV estimates based on Ozone Monitoring Instrument (OMI) satellite data have been used extensively in the last decade for many purposes. OMI is a Dutch-Finnish built wide swath push broom instrument on-board NASA's EOS-Aura satellite that was launched in 2004 and has operated since then (Levelt et al. , 2019). There have been several validation studies carried out (e.g., Tanskanen

et al. (2007), Kazadzis et al. (2009), Bernhard et al. (2015), Zempila et al. (2018), Zhang et al. (2019), Lakkala et al. (2020)) indicating a relatively good correspondence between satellite- and ground-based UV measurements. However, there are limitations and uncertainties in the satellite-based estimates that data user needs to consider. The major limitations have been linked to the conditions of highly reflecting seasonal snow/ice cover and regions of strong loading and variability in absorbing aerosols. For the latter problem, Krotkov et al. (2005) and Arola et al. (2009) proposed a correction for absorbing

aerosols exploiting monthly aerosol absorption climatology. This correction is included in the OMI surface UV data. This same correction is applied also in the operational surface UV algorithm of Copernicus Sentinel 5 precursor (TROPOMI) (Lindfors et al. , 2018). Therefore, although the following discussion is mainly related to the OMI surface UV algorithm, it is also worth emphasizing that these issues and details in the aerosol corrections are applicable to other surface-UV algorithms as well, e.g., to the current TROPOMI UV algorithm (Lindfors et al. , 2018).

The OMI surface UV algorithm calculates first the clear-sky UV and then accounts for the attenuation by clouds/non-absorbing aerosols and absorbing aerosols by separate correction factors. When estimating the total aerosol effect in surface





UV irradiance, there are two major classes of aerosols to be considered: 1) aerosols that only scatter UV radiation and 2) aerosols that both scatter and absorb UV radiation. The former effect is included in the measured Lambertian Equivalent reflectivity (LER) scene reflectivity, so the scattering aerosol attenuation is estimated through OMI cloud correction scheme,

approximating the aerosol reflectivity by clouds of equivalent reflectivity. On the other hand, for the correction of absorbing aerosols, ancillary information is needed and currently a monthly climatology is used to obtain the necessary information for the attenuation by absorbing aerosols. Aerosols that absorb UV radiation attenuate the surface UV radiation more strongly than non-absorbing aerosols of the same optical depth. Moreover, since these aerosols also attenuate the outgoing satellite-measured radiance, the cloud correction algorithm that treats these aerosols as purely scattering underestimates their optical

depth causing overestimation of UV irradiance (Krotkov et al. , 1998). Therefore, it is a complex and difficult task to properly estimate the overall total effect by scattering and absorbing aerosols. In this study, however, we show that there is still room for an improvement to better account for the solar zenith angle dependence and non-linearity in the absorbing aerosol attenuation and we propose a modified correction scheme. And more specifically, the innovation is to explicitly include the solar zenith angle dependence and moreover have a functional form for the correction, which can better account for the true non-linearity

in the correction over wider range in the aerosol absorption optical depth, if compared to the currently operational correction.

This paper is organized as follows. Section 2.1 first introduces the background and principle in the correction for absorbing aerosols, which is currently operational in OMI and TROPOMI surface-UV algorithms. Then in the Section 2.2, the radiative transfer simulations and assumptions are described, followed then by the specific details used to derive a new correction for absorbing aerosols in the Section 2.3. In the Section 3 some examples of differences between new proposed correction and the

currently operational one are shown as global maps at noon-time conditions. Finally, Section 4 summarizes our study and main findings.

## 2    Methodology

### 2.1    Principle of the correction for absorbing aerosols

The OMI surface UV algorithm first estimates the clear–sky surface irradiance using the OMI-measured total column ozone,

climatological surface albedo (Tanskanen, A. , 2004), elevation above sea level, solar zenith angle (SZA), and latitude–dependent climatological ozone and temperatures profiles. In the next step, the clear–sky irradiance is multiplied by cloud correction $C_c$, which also accounts for scattering aerosols and also by correction factor for aerosol absorption. If we denote the clear-sky UV as $UV_{clear}$, and the correction factors for cloud/scattering aerosol and absorbing aerosol as $C_c$ and $C_a$, respectively, we can write the equation for the cloudy sky $UV_{cloud}$ as:

$$UV_{cloud} = C_c * C_a * UV_{clear} \tag{1}$$

The wavelength dependence for all terms in Eq. (1) was omitted for clarity. In the OMI surface UV algorithm, effective cloud and scattering aerosol optical depth (COD) is retrieved using 360nm channel reflectance. Although COD is assumed spectrally





constant, the $C_c$ factor has characteristic spectral dependence with broad maximum at 330-340nm due to interaction between Rayleigh scattering and cloud layer, decreasing at shorter UVB wavelengths due to ozone absorption. The cloud correction of OMI surface UV is based on radiative transfer calculations for a homogeneous, plane-parallel water-cloud model embedded in a scattering molecular atmosphere with ozone absorption (Krotkov et al. , 2001). The cloud optical depth, which is assumed to be spectrally constant with the angular scattering corresponding to the C1-cloud model (Deirmendjian , 1969), is derived from OMI-measured 360 nm radiance assuming aerosol-free atmosphere.

Estimates of surface UV fluxes are further corrected for the effects of absorbing aerosols by applying an additional correction factor $C_a$ as described by Arola et al. (2009). This correction factor is based on monthly aerosol climatology of aerosol absorption optical depth (AAOD) by Kinne et al. (2013) at 1x1 degree latitude-longitude resolution. Different correction factors are estimated for each wavelength of the surface UV product, using wavelength dependent aerosol absorption optical depth (AAOD). In the following, however, we use 360nm to derive the new correction, since it is then consistent with the wavelength of reflectance used for the scattering aerosol correction, as described in the above paragraph.

There is a new version of the aerosol climatology available and published recently (Kinne , 2019) and we plan to include it in the next operational version OMI surface UV algorithm. The main parameters required for correction are the aerosol optical depth, $\tau_{aer}$ and the aerosol single scattering albedo, $\omega$. These are needed to produce the global fields for aerosol absorption optical depth, $\tau_{abs}$ [$\tau_{abs} = \tau_{aer} * (1 - \omega)$] used in the parameterization proposed by Krotkov et al. (2005); Arola et al. (2009):

$$C_a^{OMI} = \frac{1}{1 + K * \tau_{abs}}, \tag{2}$$

where $C_a^{OMI}$ is the post-correction factor, to account for absorbing aerosols in Equation 1. It is denoted here additionally by word "OMI" to distinguish it from the new parameterization developed in this study, which in turn is denoted hereafter as $C_a^{NEW}$. The part $1 + K * \tau_{abs}$ of this equation, with a slope term $K$, describes the overestimation factor of satellite-based UV due to aerosol absorption, i.e., ($UV_{clear} * C_c / UV_{cloud}$).

Previous studies (Arola et al. , 2005; Krotkov et al. , 2005) acknowledged that the slope $K$ depends on solar zenith angle (SZA) and $\tau_{abs}$ but neglected these dependencies using average value of $K = 3$ with the Equation 2 as suggested by Krotkov et al. (2005) and Arola et al. (2009). This choice was mainly based on limited validation results that included ground UV measuring stations with moderate level of absorbing aerosols. However, in this study we revisited this assumption and developed a modified algorithm to account for both SZA and $\tau_{abs}$ dependency in the absorbing aerosol correction. This was considered as an important step to enhance the applicability of the correction globally, also in regions of high seasonal biomass burning, for instance.

The new correction scheme was developed with the aid of radiative transfer (RT) simulations with LibRadtran RT package (Emde et al. , 2016) and compared to the current simpler correction. In the following sections, these simulations are described and explained.





## 2.2 Radiative transfer simulations to build up the new correction

To establish a new correction for absorbing aerosols, which accounts for both SZA and AAOD dependencies, we carried out a comprehensive set of RT simulations. Since the aerosol correction is divided into two separate terms in the satellite-UV algorithm (corrections for aerosol scattering by $C_c$ and for aerosol absorption by $C_a$), we needed to estimate "OMI-like" COD and thus estimate $C_c$ that the satellite-UV algorithm would assume for any given aerosol conditions. In addition, "true correction" of full aerosol effect (effect of both aerosol scattering and absorption), $C_{true}$, was estimated as a ratio of surface UV

flux from two RT model runs: run with aerosols and $UV_{clear}$ (see Equation 1). Since our goal was to derive a new correction for absorbing aerosols, which should be directly applicable in those surface-UV algorithms that use a similar principle than OMI and TROPOMI (described by the Equation 1), the following should be emphasized. It was indeed crucial that for our $C_c$ estimation we included water clouds only, to be consistent with the scattering aerosol treatment in those algorithms. Moreover, although these algorithms do not distinguish between water clouds, haze, ice clouds and non-absorbing aerosols, sensitivity

studies have shown that for AOD of 0.5 at 360 nm for instance, the error in estimating the $C_C$ for these varying conditions through water cloud assumption is relatively small, about 1% (Krotkov et al. , 2002).

According to the Equation 1, with a perfect algorithm $C_c * C_a = C_{true}$, in which case $C_a = C_{true}/C_c$: ratio between the "true correction" and the correction if only the scattering aerosol correction was applied ($C_c$ only, as in OMI cloud correction algorithm). These RT simulated ratios of $C_a = C_{true}/C_c$ formed the basic source of information to find a suitable new

parameterization to describe the correction factor for absorbing aerosols as a function of SZA and AAOD, $C_a^{NEW}$.

The assumption in the cloud correction, $C_c$, is that it also accounts for scattering aerosols. However, scattering aerosols and cloud droplets differ by size and thus also by their scattering angular dependence, cloud droplets being more forward-scattering. This means that, for instance, for a scene of purely scattering aerosols in cloud-free conditions OMI-retrieved effective COD should be larger than the true AOD. To put this slightly differently: in cloud-free conditions with scattering aerosols of a given

aerosol optical depth (AOD), the effective cloud optical depth must be larger than the AOD to cause the same reflectance at top of the atmosphere. In general, this difference between COD and true AOD depends on the aerosol optical properties, most strongly on single scattering albedo (SSA). In order to properly estimate $C_c$, we created a following simulation set up. First, we simulated top-of-atmosphere (TOA) radiance measurements at 360nm for clear-sky (no clouds) atmosphere that nadir-looking satellite instrument would measure from varying aerosol conditions (so varying AOD and SSA, and assuming a constant value

of 0.7 for the aerosol asymmetry parameter at 360nm). Then in the second step, radiancies were similarly simulated, but for the case of aerosol-free atmosphere with clouds (and varying COD in this case). The latter case corresponds to the OMI cloud correction, which assumes homogeneous C1 cloud model without aerosols. Therefore, we can find effective COD that OMI would retrieve for a given cloud-free scene including aerosols with varying AOD and SSA. We can then estimate $C_c$ as a ratio of simulated surface UV flux with "OMI-like" COD and $UV_{clear}$. This correction factor then reduces the surface UV to the

extent that is due to the aerosol scattering.

Similarly to OMI, we assume only water clouds in our simulations. We also evaluated the influence of the satellite viewing zenith angle (VZA) on the correction factor, but found only a minor influence. In these nadir view simulations, we then only



varied solar zenith angle (SZA) and used fixed atmospheric profile of AFGL (Air Force Geophysical Laboratory) mid-latitude summer from the LibRadtran library and disort with 16 streams as the RT solver. The cloud layer was placed between 2 and
155 4km and default aerosol profile of LibRadtran was used, therefore placing the main fraction of aerosols close to surface and in the boundary layer. In addition, fixed surface albedo of 0.03 was assumed. The main goal in our work here was to develop a new correction that properly accounts for both SZA and AAOD dependence, while preserving the level of sophistication regarding the secondary factors similar to the previous method.

To develop a new correction for absorbing aerosols, the actual total aerosol effect on surface UV needs to be taken into
160 account. In other words, two-fold effect by absorbing aerosols needs to be considered, to account for both the possible influence of absorbing aerosols in the satellite-measured TOA radiance (thus for the possible low bias in COD) and also for the impact of aerosol absorption in atmospheric attenuation of surface UV. For this reason, the impact in total transmission is assessed by $C_{true}/C_c$. This is the quantity, for which the new correction for absorbing aerosols, is to be developed. However, it is illustrative to show first the ratio of true correction and the current correction for absorbing aerosols, which was suggested in
Arola et al. (2009), $C_a^{OMI}$. The Figure 1 shows the error involved in the current correction, i.e the ratio of $C_{true}/(C_c * C_a^{OMI})$ as a function of AOD and SSA, where $C_a^{OMI}$ was calculated according to the Equation 2 with the constant $K$ of 3.

In these simulations, there were a small number of cases when negative COD was retrieved. In other words, these are cases when the aerosol absorption was so strong, relative to the aerosol scattering, that it diminished the TOA reflectance to such a low level that the signal by aerosol scattering vanished. These few cases were in the left bottom corner of the plot, when both
AOD and SSA were very low. In the Figure 1 these cases are now included as zero COD and thus with $C_c$ of 1. Moreover, it is to be noted that the range covers very high cases of aerosol absorption, up to AAOD of about 0.35. Those absorption levels do not occur often but are nonetheless possible in some regions during the seasons of biomass burning or dust aerosols, for instance.

As mentioned above, in case of ideal perfect algorithm $C_a * C_c$ would equal $C_{true}$, thus with such an algorithm the ratios
shown in the Figure 1 would be always one. However, since $C_a^{OMI}$ is a rather simple parameterization, Figure 1 illustrates both the apparent SZA and AAOD dependency of "true correction", if compared to the current operational one. On the other hand, these results also confirm that for purely scattering aerosols (SSA=1), the current version of the algorithm properly accounts for the overall aerosol effect for all SZA. In the current correction, there is no SZA dependence and the constant slope of 3 has been estimated using a data set including a range of SZA values, and as a result it over-corrects (under-corrects) at low SZA (at
high SZA). This SZA dependency is the reason why the ratios shown are mostly larger than 1 in the upper plot, while they are lower than 1 in the lower plot when SZA is higher. There is also another major influence to be considered when interpreting the results in the Figure 1, that is the AAOD dependence in the correction, which in turn includes two types of effects. First, absorbing aerosols cause attenuation in the surface irradiance. Second, the satellite-measured reflectance is decreased due to the aerosol absorption, which leads to the underestimation of COD and $C_c$. Both effects are increasing with increasing AAOD
and are not fully accounted for by the current correction (Equation 2) in higher AAOD. In the conditions of SZA of 20° (lower plot), there is an overall over-correction due to the SZA effect by the Equation 2, however when AAOD is increasing (AOD increasing and/or SSA decreasing), there is simultaneously increasing under-correction due to the true non-linear AAOD





dependence, and as a result the overall effect maximum maximizes around AOD of 1 and diminishes then for larger AOD. On
the other hand, with the case of SZA of 60°, both the neglected SZA dependence and AAOD influence are responsible for
under-correction, as can be interpreted also from the lower plot. Both SZA and AAOD impacts are further illustrated later by
Figure 2, when new correction is compared against current correction.

### 2.3 Derivation of the new enhanced correction for absorbing aerosols

Our RT simulations covered a wide range of SZAs from 0 to 80°, as well as a broad range of AOD and SSA, as discussed
above. The ultimate objective was to establish a new correction, $C_{true}/C_c$ as a function of AAOD and SZA. The current
operational formula (Equation 2) is a function of AAOD only. Moreover, the denominator $(1+K*AAOD)$ is linear with respect
to AAOD, while in our analysis we found that it does not properly describe the non-linearity of the actual correction factor
with respect to AAOD. Moreover, clear SZA dependency exists in the correction factor that the earlier approach did not take
into account. Therefore, our goal was two-fold: to keep the formula still as simple as possible, but to account for AAOD and
SZA dependencies.The final parameterization was found after an extensive search for the most appropriate form, so essentially
by an "trial and error" approach, resulting in the following equation:

$$C_a^{NEW} = 1 + c1*f + c2*f^2 + c3*f^3, \tag{3}$$

where $f$ describes the SZA dependent part in the correction factor, for which the suitable form was the following: f=(1.27+
$sin(SZA))*\tau_{abs}$. This formula provides the best fit for the overall range of SZA, AOD, and SSA of our simulations for the
true correction factor, $C_{true}/C_c$. In addition, the following constants were found to best describe the correction factor for these
205 various conditions of AAOD and SZA: $c1$=-1.43, $c2$=1.20, $c3$=-0.56.

The upper plot of the Figure 2 shows the performance of this proposed new method at two SZA values (20 and 60°) and the
current operational OMI correction, which does not depend on SZA. The red and green solid lines show the corrections that
are based on the suggested formula (Equation 3) at SZA of 20 and 60°, respectively. The red and green points, on the other
hand, show the RT simulations of $C_{true}/C_c$ (as in the Figure 1) as a function of AAOD. The current operational correction of
210 Arola et al. (2009) is shown by blue line. Similar features are apparent in this plot that were already discussed above, that is
the current algorithm is overestimating the absorption effect at low SZA values and underestimates at high enough SZA. Also
the true AAOD dependency is better captured by the new approach. The lower plot, on the other hand, shows the ratio of the
new proposed correction and currently operational correction for absorbing aerosols at these two SZA values, indicating that
the difference is most pronounced at low SZA conditions and with substantial aerosol absorption.

### 3 Comparison of new correction against the previous version

It is possible that one would interpret the results in the Figure 1 so that the differences between new and current aerosol
corrections do not appear significant. Indeed, the differences are in the range of ±5% for realistic conditions. They can be larger





in rather extreme SZA, for instance, but then the UV intensities themselves are small and the correction factor itself becomes less relevant. Also, the differences can be larger for some exceptionally high AAOD levels that are not unusual seasonally (e.g., biomass burning events in South-America or South-Africa). However, it is emphasized that these are systematic errors (biases). For instance, the current correction overestimates the absorption effect in the noontime UV index (thus underestimates the surface UV) in regions where noontime SZA values are below 40-50°. Therefore, it is of importance to correct also for these systematic AAOD and SZA effects.

To illustrate the potential impact of the new correction in the global UV estimates, we applied the current and new proposed correction for global fields of AAOD from the aerosol climatology currently used in OMI UV algorithm (Kinne et al. , 2013). In the following Figures 3 and 4, we show the ratio of the corrections at noon time and for two example months of January and June. It is obvious that the difference is in the range of ±5%. It illustrates how systematic over- and underestimation of absorbing aerosol influence can be reduced by the new proposed algorithm, which is planned to be included in the new OMI UV re-processing, planned for early 2021. The relatively sharp change from over- to under-correction in the current OMI correction close to Sahel region, where there is a very strong spatial gradient in $C_a^{OMI}$ and thus in AAOD, is an interesting spatial feature to demonstrate how both AAOD and SZA indeed influence the $C_a^{NEW}$.

In addition to these two months shown, other months were investigated (not shown). For instance in September and October during the biomass burning season both in South-America and South-Africa, when climatological AAOD levels are quite high, similar differences of ±5% were observed when comparing the new correction and the current operational correction for absorbing aerosols in the OMI surface UV algorithm. However, since now these examples were produced by using monthly climatology, it is obvious that the impact would be larger in episodic cases of higher true AAOD. Moreover, as illustrated by the Figure 2 above, the influence can be also larger, in particular for cases of higher SZA than shown here at local solar noon.

It can be also concluded from these global maps that it will be likely challenging, if not entirely impossible, to see and confirm this improved performance through the possible future validation studies against ground-based UV measurements for two reasons. First, the differences are largest in those regions where there are no ground-based UV measurements available (dust-belt or regions with strong seasonal biomass burning), unfortunately. Second, even if the ground-based UV measurements were available, the differences due to the different correction versions of absorbing aerosols are still likely smaller than the typical uncertainty levels in ground-based UV measurements. However, it is re-stressed that the new correction accounts for systematic effects by SZA and AAOD, as confirmed by our radiative transfer simulations, and thus should be considered in the improved algorithm.

## 4 Conclusions

Satellite estimates of surface UV irradiance have been available since 1978 from TOMS UV spectrometer and continued with OMI instrument and most recently from TROPOMI. Initially, in these algorithms no correction for aerosol absorption was included, while in Arola et al. (2009) a correction for absorbing aerosols was suggested based on monthly climatology of Kinne et al. (2013). This correction was then applied in OMI algorithm and later also in the TROPOMI UV algorithm



(Lindfors et al. , 2018). This correction was to large extent based on rather limited ground-based data not covering a very large variations of SZA and AAOD. Although, the impact by SZA and AAOD was acknowledged, simpler approach was considered sufficient. In this study we revisited this formulation for the correction, as an objective to find a suitable modifications for the correction to better account for various atmospheric conditions. The motivation was the following: although the differences
between new proposed and current operational corrections for absorbing aerosols are not very large, they are systematic and should therefore be taken into account to improve the accuracy of the surface UV products from satellite measurements.

The new correction was derived using RT simulations of varying conditions of aerosols and solar zenith angles and a new correction is suggested. We also estimated the potential impact in future satellite UV products, after this new correction is implemented, showing a typical differences of $\pm5\%$. This new correction is equally easy to implement and replace the current
correction in OMI and TROPOMI, essentially the only new information to include is SZA, since the earlier one was already based on AAOD.

*Code availability.*   The code to reproduce the results is available from the corresponding author on request.

*Author contributions.*   A. Arola (AA) designed the study and conducted radiative transfer simulations. AA led the analysis and writing of the manuscript with significant contributions from all authors

*Competing interests.*   The authors declare that they have no conflict of interest.



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

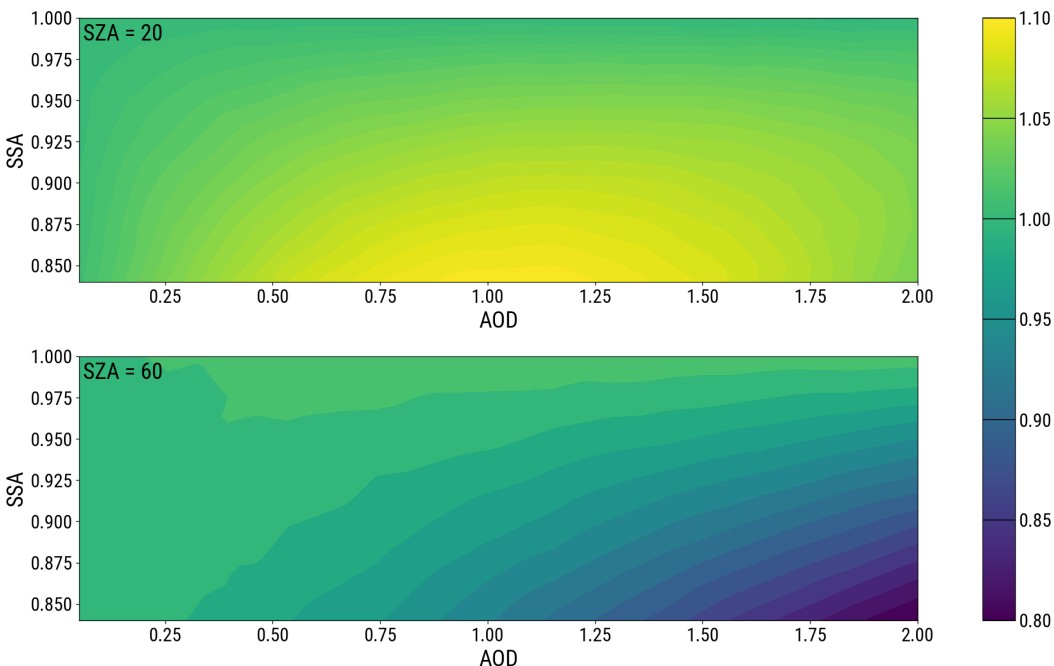

**Figure 1.** Ratio between the "true correction" and the correction of current operational OMI algorithm, $C_{true}/(C_c * C_a^{OMI})$, as a function of aerosol optical depth (AOD) and single scattering albedo (SSA). Two solar zenith angles are shown: $20°$ and $60°$ in the upper and lower plots, respectively.



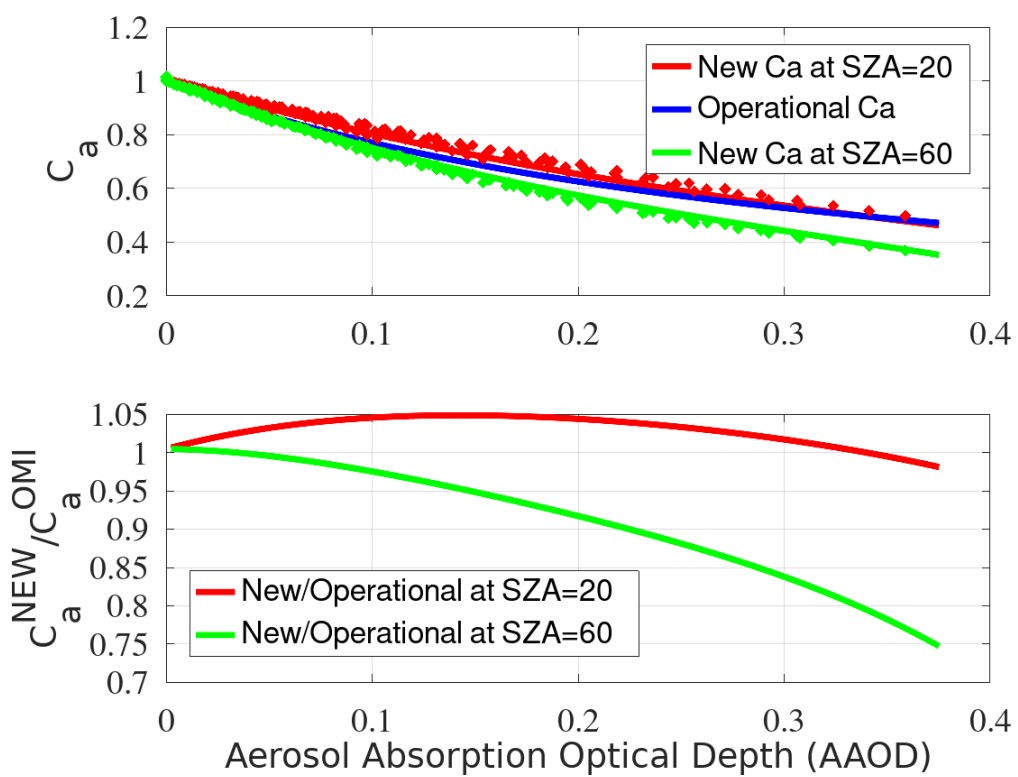

**Figure 2.** Different corrections for absorbing aerosols, $C_a$, as a function of AAOD (in the upper plot). Two cases of new proposed correction are shown, at $20°$ and $60°$ (by red and green colors, respectively) and the current operational correction (in blue). The lower plot shows the ratio of the new proposed correction and currently operational correction for absorbing aerosols, $C_a^{NEW}/C_a^{OMI}$), at these same two SZA values.

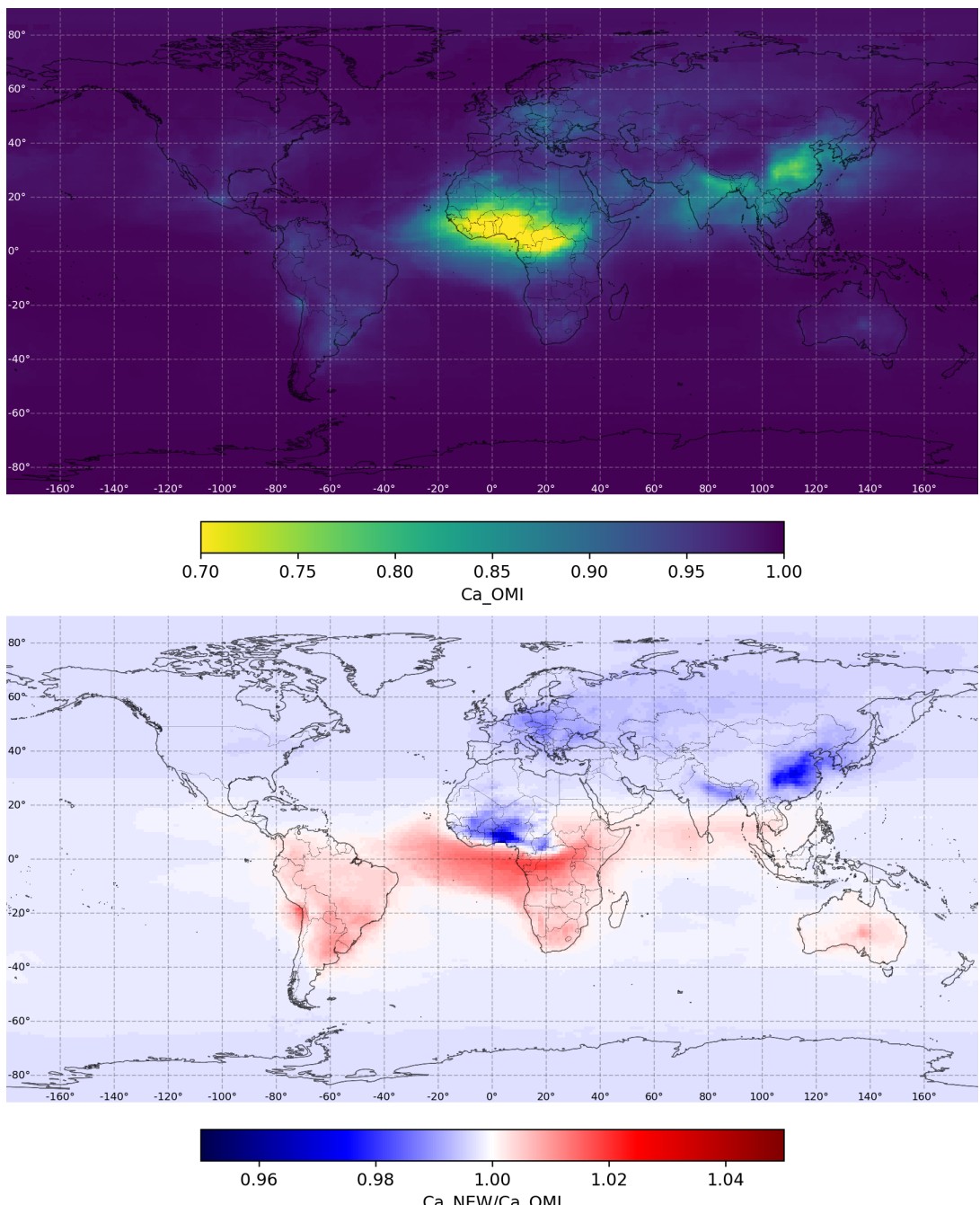

**Figure 3.** The current operational OMI correction for absorbing aerosols, $C_a$ (upper panel). The ratio of the new proposed correction and current operational correction for absorbing aerosols, $C_a^{NEW}/C_a^{OMI}$ (lower panel). Solar zenith angle corresponds to the noon time conditions on January 15th and aerosol climatology of January is used.





**Figure 4.** Same as Figure 3, but both SZA and aerosol climatology corresponds to June conditions