# Peer review of "Rethinking the correction for absorbing aerosols in the satellite-based surface UV products."

_Atmospheric Measurement Techniques, 2021_

## Author Comment (AC1)

We would like to first express our thanks to the REFEREE #2 for his/her constructive comments. The point-by-point responses are below after each reviewer's point. The reviewer points are in bold.

**The manuscript introduces a simple way to improve the aerosol effect accuracy into the satellite-based surface UV products, and more specifically to the OMI and TROPOMI UV algorithms. The innovation relies on the solar zenith angle and the non-linearity incorporation into the new aerosol absorption correction scheme. The proposed correction and the whole approach will be useful for the related community, since it will make the direct UV satellite retrievals more reliable under cloudless conditions.**

**General comments:**

**The title of the paper has to address the specific satellites that this correction is valid. In the current form implies that is can be applied to all UV-related satellites.**

The title in the revised version has been modified as suggested.

**The authors mention the UV absorbing aerosol index (UVAI) and the aerosol absorption optical depth (AAOD) but in the analysis the focus is limited to the AAOD. A brief description of both and a related sensitivity analysis followed by the corresponding references could be added in the introduction.**

The UVAI is qualitative parameter, which depends not only on AAOD, but also on aerosol height or other conditions, such as sun glint. It was used in the original TOMS UV algorithm, as proxy for AAOD, but was not adapted to OMI or TROPOMI. This is why, for the sake of clarity, UVAI mention (which was only once in the abstract anyway) is now excluded in the revised manuscript.

**The potential impact of the new correction was tested against the current correction for the OMI and TROPOMI UV algorithm in terms of global fields of climatological AAOD showing a difference of ±5%. This difference seems low but it is not for the UV impacts on human health. A small paragraph of the real impact of this ±5% has to be added in the conclusions section.**

Thanks. We feel that the discussion of real health impacts by ±5% in surface UV goes beyond our expertise. Moreover, it is good to remember that most often the long-term accumulated exposure matters, for which purpose satellite data exhibit additional challenges, like description of diurnal cloudiness, for instance.

**The whole analysis is based on the existing AAOD climatology since this is the way that OMI represents the aerosol absorption effect. However, a more generalized result and conclusion will be of value for the UV community in order to quantify the limitations and gaps of the current and the proposed correction and ideally to describe which is the optimum representation approach of the absorbing aerosol loads.**

First, to our knowledge the most advanced operational surface UV algorithms (TROPOMI, OMI, GOME-2) all use aerosol climatology to account for absorbing aerosols. Climatology as an ancillary source of information has been considered as the best option. For instance, from a separate OMI algorithm AAOD would have been available, and it has been seriously considered to replace AAOD from climatology. However, this data are only available in cloud-free conditions and have significant enough uncertainties, so in the end it has been considered meaningful not to switch from climatology to the use of satellite-based AAOD instead. Therefore, we do consider the use of climatology as the "optimum representation approach of the absorbing aerosol loads", but we are also among the first to admit that it is not the perfect solution.

**Is this study intended to upgrade only the OMI approach or targets to a more general UV estimation improvement that applies to more instruments and techniques? In the conclusions section a paragraph about the necessity for a more generalized approach for all UV satellite instruments could be added in order to strengthen the innovation of this publication and to mention the need for new ones in this field.**

The purpose of our study was to develop an approach to upgrade OMI- and TROPOMI-like algorithms, which we consider as most advanced operational algorithms, in addition to GOME-2 (Kujanpää and Kalakoski 2015). GOME-2 algorithm uses slightly different approach to account for absorbing aerosols, for which our correction scheme is not directly applicable. In their approach a globally constant single scattering albedo is used (the most crucial piece of information to determine the attenuation caused by absorbing aerosols, explicitly incorporated through AAOD in our case), so that algorithm clearly has its own avenues and room to make improvements.

**Corrections:**

**Line 35: Add a reference.**

Reference is added into the revised manuscript.

**Line 38: The continuous global monitoring without time gaps requires geostationary satellites and a harmonization between their retrievals and their implicit radiative transfer and correction schemes.**

We agree. However, surface UV estimates from meteorological geostationary satellites have not been possible, while it is indeed an exciting possibility for the new generation of geostationary spectrometers, e.g., GEMS, TEMPO and Sentinel 4 missions. Therefore, the statement emphasizing the long-term existing data set was left unchanged. However, we have added the following sentence in this section:

New generation of geostationary UV-visible spectrometers, such as GEMS [Kim et al., 2020], TEMPO [Chance et al., 2019] and European Sentinel 4 will offer an exciting new possibility of hourly surface UV estimates accounting for diurnal changes in clouds, aerosols and ozone absorption.

**Lines 52-54: Needs a connection also with other spectrometers, e.g. OMPS and GOME-2.**

As mentioned above, GOME-2 is having a different approach (while OMPS UV product does not exist).

**Line 59: The LER was first mentioned in the Abstract, so no need to repeat the full name.**

AMT guidelines say the following about the abbreviations: "They need to be defined in the abstract and then again at the first instance in the rest of the text."

**Line 60: Add a reference.**

Krotkov et al. 1998 added.

**Lines 62-63: This is a repetition from the Abstract. You can analyze more here with the corresponding references.**

The abstract is considered as an individual summary, separate from the other sections of the manuscript. Therefore, these fundamental points were repeated also here in the main text (in the introduction).

**Lines 87 and 91: Probably you have to change the abbreviation of the scattering aerosol optical depth in order to avoid confusion with the cloud optical depth (or rename the later into cloud optical thickness).**

OMI and TROPOMI UV algorithms indeed determine combined cloud and aerosol optical depth (COD) from the reflected radiance at 360nm, which only approximates the true scattering aerosol optical depth for cloud-free scenes. Therefore, the terminology was left unchanged.

**Lines 96 and 98: The AAOD abbreviation was first introduced into the Abstract, so no need to repeat the full name.**

AMT guidelines say the following about the abbreviations: "They need to be defined in the abstract and then again at the first instance in the rest of the text."

**Line 101: A comparison under a variety of conditions and exceptions has to be mentioned here and studied in this or in a next publication in order the new correction to provide a tangible improvement to the updated UV retrievals.**

As we mentioned in the manuscript, we plan to update the aerosol climatology in the future re-processing of OMI surface UV records. However, the scope in this study was to improve the actual algorithm, which can use any reasonable AAOD climatology. Clearly the influence of climatology itself is out of scope of this study. Nevertheless, for the interest of our reviewer we spent some time to compare MAC V2 and V1 aerosol climatologies and the corresponding correction factor, Ca. Overall, the differences were relatively small, but some localized larger differences were apparent. These regional patterns were most typically in Sahara/Sahel region and in South-East China and could reach up to 10% (both positive and negative differences were found, depending on the season and region).

**Lines 106-107: A list of abbreviations could be helpful for the readers.**

We hope our new figure of the RT modeling exercise (Figure 1 in the revised manuscript) would make it easier to follow not only the RT approach itself, but also the meaning of different abbreviations.

**Lines 109-110: Introduce the abbreviation of solar zenith angle in the Abstract.**

Done

**Line 114: Add a reference.**

Reference added in the revised manuscript.

**Line 139: Add a reference.**

Reference added in the revised manuscript.

**Line 142: Add a reference.**

Reference added in the revised manuscript.

**Lines 142-146: This description could be supported by a flowchart plot, since it forms the base for the proposed new correction.**

We agree and thank reviewer for this excellent suggestion. We added an illustration of different steps in the RT simulations (Figure 1 in the revised version). We hope it clearly clarifies our approach.

**Line 151: Why only water clouds?**

We considered only water cloud model, because the OMI/TROPOMI cloud correction algorithm derives effective COD using classic C1 water cloud model [Deirmendjian, 1969], which optical properties are well-known, and it is widely used in RT modeling and satellite retrieval communities. In this study we focused only on the absorbing aerosol correction, but we had to be fully consistent with the current OMI/TROPOMI algorithm. Otherwise, we would have ended up working on a complete overhaul of the entire algorithm, while for the cloud correction (Cc) algorithm part we do not see particularly valid reasons to change it.

**Line 152: Add a brief quantified value for the mentioned minor influence.**

This is done in the revised version of the manuscript.

**Line 153: Keep only the SZA and remove the full name.**

Done as suggested.

**Lines 154-155: Is the simulated cloud layer representative globally or needs a "rethinking" next years in order to provide more accurate results across the globe climatologically? Is this factor secondary to the overall UV estimation levels or to the focus of this study into the aerosol uncertainty? A brief description of the parameters that affect the UV and the corresponding order of magnitude of the impact could be added in this paragraph or in the last paragraph before the conclusions section (i.e. lines 238-245).**

Here we want to repeat our earlier comment that we do not see currently any clear reasoning why we would change our cloud correction algorithm. The detailed description of the parameters that affect the surface UV irradiance was provided in our previous algorithmic papers, which we cite, e.g., [Krotkov et al., 1998, 2001, 2002; Tanskanen Lindfors, et al., 2018]

**Lines 159-162: Needs more discussion.**

These few sentences are clarified in the revised manuscript.

**Line 169: Add a reference.**

There is no appropriate reference.

**Lines 169-170: Is this a critical assumption and which is the impact into the global scale in terms of percentage of potential similar cases? A literature analysis could be helpful.**

These are unrealistic cases (very low AOD and extremely low SSA). This is now mentioned in the revised manuscript.

**Lines 172-173: Add a reference and some relevant numbers.**

This statement was excluded in the revised manuscript, since the maximum AAOD was likely too large to be found in real world. On the other hand, it was of interest to show the behavior over this very wide range, while with the actual climatology (Figures 4 and 5 in the revised version) the reader can see the influence in fully realistic cases (and in real world they would be even larger in episodic cases that climatology cannot include).

**Lines 178-180: Provide a number about the current over-correction (under-correction) and the expected improvement with the new correction.**

These numbers are elaborated elsewhere in the text already few times, so we considered it as a repetition if mentioned here as well.

**Line 198: Is it possible to add the expected difference in the calculation time? The addition of the SZA and the AAOD dependencies could result a more complicated calculation scheme in terms of calculation time (but still simple as a formula).**

This should not make computations slower, since both methods are post-corrections done with a prescribed analytical equation.

**Line 204-205: Can you add a sensitivity plot for the constants (as a supplement material)?**

The constants were re-estimated by slightly different method (resulting in slightly modified constants), which also provided uncertainty estimates. These uncertainty estimates are now reported in the revised manuscript.

**Line 232: The additional maps can be added as well into a supplement document. It will be interesting for the readers to see the correction differences during all months.**

All the other months were included in the supplement, as suggested by the reviewer.

**REFERENCES**

Chance, K., X. Liu, C. Chan Miller, G. González Abad, G. Huang, C. Nowlan, A. Souri, R. Suleiman, K. Sun, H. Wang, L. Zhu, P. Zoogman, J. Al-Saadi, J-C. Antuña-Marrero, J. Carr, R. Chatfield, M. Chin, R. Cohen, D. Edwards, J. Fishman, D. Flittner, J. Geddes, M. Grutter, J.R. Herman, D.J. Jacob, S. Janz, J. Joiner, J. Kim, N.A. Krotkov, B. Lefer, R.V. Martin, O.L. Mayol-Bracero, A. Naeger, M. Newchurch, G.G. Pfister, K. Pickering, R.B. Pierce, C. Rivera Cádenas, A. Saiz-Lopez, W. Simpson, E. Spinei, R.J D. Spurr, J.J. Szykman, O. Torres, and J. Wang, TEMPO Green Paper: Chemistry, physics, and meteorology experiments with the Tropospheric Emissions: Monitoring of Pollution instrument, Proc. SPIE 11151, Sensors, Systems, and Next-Generation Satellites XXIII, 111510B, 2019. https://doi.org/10.1117/12.2534883

Kim, J., Jeong, U., Ahn, M.-H., Kim, J.H., Park, R.J., Lee, Hanlim, Song, C.H., Choi, Y.-S.,

Lee, K.-H., Yoo, J.-M., Jeong, M.-J., Park, S.K., Lee, K.-M., Song, C.-K., Kim, Sang-Woo, Kim, Y., Kim, Si-Wan, Kim, M., Go, S., Liu, X., Chance, K., Miller, C.C., Al-Saadi,J., Veihelmann, B., Bhartia, P.K., Torres, O., Abad, G.G., Haffner, D.P., Ko, D.H., Lee,S.H., Woo, J.-H., Chong, H., Park, S.S., Nicks, D., Choi, W.J., Moon, K.-J., Cho, A.,Yoon, J., Kim, S., Hong, H., Lee, K., Lee, Hana, Lee, S., Choi, M., Veefkind, P., Levelt,P., Edwards, D.P., Kang, M., Eo, M., Bak, J., Baek, K., Kwon, H.-A., Yang, J., Park, J.,Han, K.M., Kim, B.-R., Shin, H.-W., Choi, H., Lee, E., Chong, J., Cha, Y., Koo, J.-H.,Irie, H., Hayashida, S., Kasai, Y., Kanaya, Y., Liu, C., Lin, J., Crawford, J.H.,Newchurch, G.R.C.M.J., Lefer, B.L., Herman, J.R., Swap, R.J., Lau, A.K.H., Kurosu,T.P., Jaross, G., Ahlers, B., Dobber, M., McElroy, C.T., Choi, Y.,  New Era of Air Quality Monitoring from Space: Geostationary Environment Monitoring Spectrometer (GEMS), Bull. Amer. Met. Soc., 2020, https://doi.org/10.1175/BAMS-D-18-0013.1

Kujanpää, J. and Kalakoski, N.: Operational surface UV radiation product from GOME-2 and AVHRR/3 data, Atmos. Meas. Tech., 8, 4399–4414, https://doi.org/10.5194/amt-8-4399-2015, 2015.

---

## Author Comment (AC2)

We would like to first express our thanks to the REFEREE #1 for his/her constructive comments. The point-by-point responses are below after each reviewer's point. The reviewer points are in bold.

**General Comments**

The manuscript proposes an improved algorithm to better account for the impacts of absorbing aerosols in OMI/TROPOMI surface UV products. This proposed scheme can be easily implemented to correct the systematic effects caused by SZA and AAOD in the current OMI operational surface UV algorithm. This paper is interesting and well written.

**1.** It seems that the correction algorithm developed in this work is targeting OMI and TROPOMI satellites, it would be helpful to add this information in the title of the manuscript.

The title in the revised manuscript has been modified.

**2. How applicable is this proposed algorithm to be used in other satellite derived surface UV products? It would be helpful to add some comments on this.**

The correction is targeting OMI and TROPOMI satellites as its basic features are linked with the OMI and TROPOMI instruments measuring the sum of cloud reflected and aerosol backscattered spectral radiance. The correction could be applicable for future satellite based instruments having the same measurement principles. This has been further stressed in the revised manuscript.

3. Line 64-67: Here could discuss a little bit on what the surface UV estimates would be without accounting for the SZA dependence and the non-linearity in the correction scheme, such as whether they would be systematically overestimated or underestimated compared to the current operational algorithm.

There can be both over- and underestimation, depending on the SZA (and AAOD as well). This was discussed in the manuscript related to the Figure 3, because the most natural place for this discussion seemed to be in the context of the results in illustrated by that plot.

**4. Line 98-99: how are the new aerosol climatology data different from the current climatology data? How is it going to affect the results?**

As we mentioned in the manuscript, we plan to update the aerosol climatology in the future reprocessing of OMI surface UV records. For the interest of our reviewer we spent some time to compare MAC V2 and V1 aerosol climatologies and the corresponding correction factor, Ca. Overall, the differences were relatively small, but some localized larger differences were apparent. These regional patterns were most typically in Sahara/Sahel region and in South-East China and could reach up to 10% (both positive and negative differences were found, depending on the season and region).

**5. Section 2 has a lot of texts and it is hard to go through. It would be helpful for the readers to read through it if it can be better organized such as showing a flowchart of the algorithm or reorganizing some of the long paragraphs.**

A new figure (Figure 1) and equation (Equation 3) have been included in the revised manuscript, which both hopefully clarifies the revised discussion in the section 2.

**6. Line 228-229: It would be interesting to see the plots in other months.**

The new figures have been included in the appendix.

**7. Line 35: a reference is needed here.**

Reference is added into the revised manuscript (Schmalwieser et al., 2017)

**REFERENCES**

Schmalwieser AW, Gröbner J, Blumthaler M, Klotz B, De Backer H, Bolsée D, Werner R, Tomsic D, Metelka L, Eriksen P, Jepsen N, Aun M, Heikkilä A, Duprat T, Sandmann H, Weiss T, Bais A, Toth Z, Siani AM, Vaccaro L, Diémoz H, Grifoni D, Zipoli G, Lorenzetto G, Petkov BH, di Sarra AG, Massen F, Yousif C, Aculinin AA, den Outer P, Svendby T, Dahlback A, Johnsen B, Biszczuk-Jakubowska J, Krzyscin J, Henriques D, Chubarova N, Kolarž P, Mijatovic Z, Groselj D, Pribullova A, Gonzales JRM, Bilbao J, Guerrero JMV, Serrano A, Andersson S, Vuilleumier L, Webb A, O'Hagan J. UV Index monitoring in Europe. Photochem Photobiol Sci. Sep 13;16(9):1349-1370. doi: 10.1039/c7pp00178a. PMID: 28848959, 2017.